# Engineering and Assessing Cardiac Tissue Complexity

**DOI:** 10.3390/ijms22031479

**Published:** 2021-02-02

**Authors:** Karine Tadevosyan, Olalla Iglesias-García, Manuel M. Mazo, Felipe Prósper, Angel Raya

**Affiliations:** 1Regenerative Medicine Program, Bellvitge Institute for Biomedical Research (IDIBELL) and Program for Clinical Translation of Regenerative Medicine in Catalonia (P-CMRC), 08908 L’Hospitalet del Llobregat, Spain; ktadevosyan@idibell.cat; 2Center for Networked Biomedical Research on Bioengineering, Biomaterials and Nanomedicine (CIBER-BBN), 28029 Madrid, Spain; 3Regenerative Medicine Program, Cima Universidad de Navarra, Foundation for Applied Medical Research, 31008 Pamplona, Spain; mmazoveg@unav.es (M.M.M.); fprosper@unav.es (F.P.); 4IdiSNA, Navarra Institute for Health Research, 31008 Pamplona, Spain; 5Hematology and Cell Therapy Area, Clínica Universidad de Navarra, 31008 Pamplona, Spain; 6Center for Networked Biomedical Research on Cancer (CIBERONC), 28029 Madrid, Spain; 7Catalan Institution for Research and Advanced Studies (ICREA), 08010 Barcelona, Spain

**Keywords:** pluripotent stem cells, cardiomyocytes, cardiac tissue engineering, human heart, tissue maturation

## Abstract

Cardiac tissue engineering is very much in a current focus of regenerative medicine research as it represents a promising strategy for cardiac disease modelling, cardiotoxicity testing and cardiovascular repair. Advances in this field over the last two decades have enabled the generation of human engineered cardiac tissue constructs with progressively increased functional capabilities. However, reproducing tissue-like properties is still a pending issue, as constructs generated to date remain immature relative to native adult heart. Moreover, there is a high degree of heterogeneity in the methodologies used to assess the functionality and cardiac maturation state of engineered cardiac tissue constructs, which further complicates the comparison of constructs generated in different ways. Here, we present an overview of the general approaches developed to generate functional cardiac tissues, discussing the different cell sources, biomaterials, and types of engineering strategies utilized to date. Moreover, we discuss the main functional assays used to evaluate the cardiac maturation state of the constructs, both at the cellular and the tissue levels. We trust that researchers interested in developing engineered cardiac tissue constructs will find the information reviewed here useful. Furthermore, we believe that providing a unified framework for comparison will further the development of human engineered cardiac tissue constructs displaying the specific properties best suited for each particular application.

## 1. Introduction

Cardiovascular diseases (CVD) remain the main cause of morbidity and mortality worldwide despite many advances in preventive cardiology. According to 2017 estimates, CVD accounted for over 420 million cases every year and resulted in close to 18 million deaths [1]. Acute myocardial infarction (MI) and subsequent heart failure continue to have high prevalence worldwide [2] and are among the major health issues. As the human adult heart has minimal regenerative capacity [3], cardiomyocytes (CMs) that are lost during ischemic injuries are usually replaced with fibrotic scar tissue leading to the partial or total cardiac dysfunction [4]. Today the only curative option for patients with end-stage heart failure is heart transplantation, which is usually limited due to the scarcity of donor hearts [5]. Preclinical research studies have demonstrated that cell therapy can attenuate myocardial damage and reduce the progression of cardiac remodeling to heart failure [6]. However, clinical studies have failed to show significant improvements and preliminary data indicate that stem cells have the potential to enhance tissue perfusion and contractile performance [7]. Numerous studies have demonstrated that the therapeutic benefits exerted by cells are mainly attributable to the release of complementary paracrine factors and the efficacy is limited as only a small percentage of transplanted cells engrafted in the infarcted tissue [8]. Studies on animal models showed that combining cell therapy with tissue engineering techniques for the creation of cell sheets and patches, can increase stem cell survival and boost therapeutic action [9]. Therefore, tissue engineering has been considered as a potential approach for cardiac regeneration after MI. Current research in the field is, to a large extent, aimed towards the development of functional heart tissue for application in cell-based regenerative therapies [10,11], cardiac disease modeling [12,13,14,15] and drug screening [16]. In particular, the application of cardiac tissue engineering (CTE) systems for investigating mechanisms of disease has been recently reviewed [17] and will not be specifically covered here. Despite the evident progress realized to date, engineered cardiac tissues (ECT) remain immature and still differ from the adult human heart. One of the major challenges in CTE is the organization of stem-cell derived CMs into a functional tissue of large dimensions suitable for the intended therapeutic use, and that recapitulates the main physiological properties of the real heart. In this review, we describe the main approaches undertaken toward the development of ECT constructs, along with the methodologies utilized for assessing their functionality, and discuss the current state and future directions.

## 2. Cell Sources

The heart muscle is composed of electrically and mechanically connected CMs closely surrounded by endothelial cells (ECs), fibroblasts (FBs), smooth muscle cells (SMCs), pericytes, and heart resident immune cells [18]. Even though CMs constitute approximately 70–85% of the heart by volume [19], non-myocyte cells are more abundant by cell number and bear critical roles in heart homeostasis, supporting the extracellular matrix, intercellular communication, and vascular supply essential for CM survival and contraction [19,20]. Thus, the most advanced current CTE approaches aim at generating 3-dimensional (3D) constructs that incorporate CMs as well as non-myocyte cardiac cells to reproduce the myocardial niche and allow the creation of a functional mature structure (Figure 1 and Table 1).

### 2.1. Neonatal Rodent Myocytes

The first studies on cardiac engineering in the late 1990s used cells derived from neonatal rat hearts since human CMs were not readily available and could not be expanded from cardiac biopsies [60,61,62]. Moreover, while human embryonic stem cells (hESC) were first derived in 1998, robust protocols to guide cardiac differentiation took another decade to be perfected and yield human CMs with the desired purity and reproducibility [63,64,65]. On the other hand, the use of adult ventricular myocytes from both rats and mice showed low viability and proliferation rate in long-term cultures which made them useless for CTE [66,67]. Thus, by this period of time neonatal rat cardiac myocytes (NRCMs) became the preferred cell source for CTE as they could be isolated with high yield and quality and presented comparably long-term viability in culture. Therefore, NRCMs were used to generate 3D functional cardiac tissue constructs in which cells underwent progressive electromechanical maturation over time in static conditions [22,35,38,41,48,57,60,68] while culturing these 3D constructs in dynamic systems further enhanced their functionality [24,25,26,28,34,41,69,70,71,72].

### 2.2. Pluripotent Stem Cells 

Pluripotent stem cells (PSCs), including embryonic and induced pluripotent stem cells (ESC and iPSCs), show promise for CTE as they can be expanded indefinitely in vitro and differentiated into multiple cell types including CMs. Robust protocols have been developed for cardiac differentiation of PSCs allowing to obtain high yield of CMs [63,73,74]. The therapeutic potential of both human ESC- and iPSCs-derived CMs has been previously demonstrated in different animal models of MI including mice [75], rats [76,77,78], guinea pigs [79], pigs [80], and non-human primates [81]. Remarkably, Shiba et al. demonstrated in 2016 that allogeneic transplantation of iPSC-CMs can regenerate the damaged heart in a non-human primate MI model. The authors generated iPSCs from an animal homozygous for a specific major histocompatibility complex haplotype (HT4), which were then differentiated into CMs. A total of 4 × 10^8^ iPSC-CMs were injected in the infarcted heart of HT4-heterozygous primates. Cells survived and engrafted for at least 12 weeks with no evidence of immune rejection. However, some grafts remained isolated from the host tissue and the incidence of arrhythmias increased after cell transplantation, although the incidence was transient and decreased gradually over time [82].

Transplantation of ECT constructs comprising human PSC-derived CMs has been shown to enhance cell survival and engraftment, promote tissue revascularization and improve functional properties of the injured heart. The Murry laboratory generated ECT constructs using hESC-CMs or hiPSC-CMs together with human umbilical vein ECs, and mesenchymal stromal cells (MSCs) in a type I collagen hydrogel and subjected them to uniaxial mechanical loading [36]. One week after transplantation onto injured hearts of athymic rats, the ECT constructs survived and formed grafts containing a microvascular network that was perfused by the host coronary circulation. Kawamura et al. used the cell sheet method to deliver hiPSC-CMs onto infarcted swine hearts. This procedure significantly improved contractile function, promoted vascularization and attenuated left ventricular remodeling. However, poor level of engraftment was detectable 8 weeks post-transplantation [83]. Transplantation of ECT constructs from hESC-CMs in a chronic rodent MI model has been shown to enhance engraftment rate, leading to increased long-term survival, and progressive maturation of the human CMs [30]. Engineered tissues containing hiPSC-derived vascular cells (ECs and SMCs) without CMs have also been associated with significant functional improvement, reduction of infarct size, increase in neovascularization, and recruitment of endogenous cardiac progenitor cells in a MI swine model [84]. Cardiac repair has also been reported after implantation of ECT constructs comprising hiPSC-derived cells in guinea pigs [45] and pigs [85].

Representative publications on CTE, indicating the sources and types of cells used, the type and dimension of the constructs generated, the culture conditions, and the functional analyses performed. CM, cardiomyocytes; iPSC-CM, cardiomyocytes derived from induced pluripotent stem cells; ESC-CM, cardiomyocytes derived from embryonic stem cells; EC, endothelial cells; HUVEC, human umbilical vein endothelial cells; FB, fibroblasts; hcFB, human cardiac fibroblasts; SMC, smooth muscle cells; MSC, mesenchymal stromal cells; n/a, information not available.

Notably, Menasché et al. reported the first clinical use of hESC-derived cardiovascular progenitor cells in a fibrin patch of 20 cm^2^, which was delivered to the heart of a patient suffering from severe ischemic left ventricular dysfunction. The study demonstrated the feasibility of generating a clinical-grade population of hESC-derived cardiovascular progenitors and improved the patient’s functional outcome after a 3-month follow-up [86]. In a subsequent study, the group tested the safety of this approach on 6 patients with up to one year follow-up, during which the patients did not present any clinically significant arrhythmias and showed an increase in systolic function. One patient died after surgery from treatment-unrelated comorbidities and another one at 22 months due to heart failure [87]. No tumors were detected during the follow-up period, demonstrating the safety of this approach at 1 year.

### 2.3. Mesenchymal Stromal Cells

Mesenchymal stromal cells (MSCs) are multipotent cells that have the potential to differentiate into a variety of cell types like adipocytes, chondrocytes and osteocytes, limited self-renewal capacity, and low immunogenicity [88]. Despite the adipocyte tissue and bone marrow being the most common sources for MSCs [89], they can be also isolated from synovial tissue, umbilical cord and peripheral blood [90]. MSCs were considered in autologous and allogeneic therapies for cardiac injuries due to their high expansion rate and immunomodulatory properties. In particular, numerous preclinical trials have shown that MSCs have the potential to promote cardiac repair in heart injury models through paracrine effects [91,92,93,94,95]. Moreover, engineered constructs containing MSCs have also been evaluated in preclinical models of MI. Patches consisting of autologous porcine MSCs in a fibrin hydrogel were transplanted onto infarcted swine hearts and resulted in improvement in contractile function and increase in neovascularization in the patch covered area [96]. Later, human MSCs cultured in a scaffold made of decellularized human myocardium with fibrin hydrogel were transplanted in nude rat models of acute and chronic MI. The treatments showed cardiac functional improvement 4 weeks after transplantation that was associated with the release of proangiogenic factors by MSCs [97]. Moreover, researchers have found an important structural role of MSCs for the organization of cardiac tissues. In particular, co-culture of human MSCs with hPSC-derived CMs and vascular cells within ECT constructs markedly improved their self-organization, vascular network formation and stabilization [36,59]. Finally, cell heterogeneity in ECT constructs due to the presence of MSCs has also been reported to influence successful engraftment [36] and to aid in the generation of human in vitro disease models of high pathophysiological relevance [59].

### 2.4. Non-Myocytes

Within the native heart, CMs are never alone but rather organize in a complex 3D tissue with intimately coupled non-myocyte cells. Moreover, heart homeostasis and response to disease strongly depends on the interaction between non-myocytes and cardiomyocytes. It therefore follows that recapitulating the normal heart tissue organization in the laboratory would necessitate including CMs along with non-myocytes [3,98]. Liau et al. generated two types of patches containing mouse ESC-derived CMs alone, or in combination with different ranges of cardiac FBs (3%, 6% and 12%). Based on their results, patches containing only ESC-CMs failed to form a functional syncytium, while the presence of cardiac FBs allowed the generation of synchronously contractile constructs with functional properties close to the native heart [99]. Building on those findings, Zhang et al. generated cardiac patches using human ESC-derived CMs with varying purity (49–90%) together with cardiac FBs, endothelial cells, and vascular SMCs, which inadvertently arose during cardiac differentiation. Interestingly, the conduction velocity in patches linearly increased with CMs purity, and when compared to matched 2D monolayers, 3D tissues showed higher conduction velocities but not different action potential duration or maximum capture rates. Moreover, the authors found that inclusion of non-myocytes increased the survival and the maturation level of CMs [100]. Tiburcy et al. also demonstrated the critical importance of defining the non-myocyte fraction for engineering force-generating cardiac tissues. Specifically, these authors found that the ratio of 70% hPSC-derived CMs and 30% human foreskin FBs was optimal for promoting maturation at the cellular and tissue levels [51]. Many subsequent studies have confirmed the significance of including non-myocytes in engineered cardiac constructs [32,36,51,53,55,101,102,103,104].

## 3. Cardiac Tissue Engineering Systems

A wide variety of 3D ECT constructs with different shapes and sizes/thicknesses have been fabricated with the purpose of being used as model systems for drug/toxicity testing or for application in regenerative medicine strategies [45,105]. The main categories of ECT constructs are depicted in Figure 1, with representative publications listed in Table 1 and described below.

### 3.1. Cell Sheets

The cell sheet technique was first reported by Shimizu et al. in 2002 for creating a transplantable 3D cell patch [57]. Cell sheet technology, also referred to as scaffold-free system or “Cell Sheet Engineering” [57,106,107], is based on stacking monolayers (or sheets) of CMs cultured to confluency to form 3D tissue-like structures. By using cell culture surfaces coated with the temperature-responsive polymer poly(*N*-isopropylacrylamide) (PIPAAm), it is possible to readily detach intact cellular monolayers of CMs as cell sheets by lowering the temperature, without any enzymatic treatments. Overlaying these thin 2D monolayers then results in 3D cardiac constructs [108]. The benefits of this system have been analyzed in vivo in murine animal models of MI showing improvements in cell survival, cardiac function and tissue remodeling [109]. The use of cell sheets created new opportunities for in vitro tissue engineering and helped exploring new therapies and drugs for heart diseases [58,59]. Interestingly, Sekine et al. produced in vitro vascularized cardiac tissues with perfusable blood vessels by overlaying additional triple-layer cell sheets made by NRCMs cocultured with endothelial cells. Such sheets were then transplanted under the neck skin of nude rats and connected to the local vasculature. Constructs with vessel anastomoses survived and maintained their vascular structure up to two weeks after transplantation. However, the thickness of the constructs decreased over time indicating that uniform perfusion was insufficient for whole tissue survival. Moreover, no functional analyses were performed in the study to evaluate maturation at tissue level [58]. More recently, Kawatou et al. developed an in vitro drug-induced Torsade de Pointes arrhythmia model using 3D cardiac tissue sheets. Importantly, the authors showed the importance of using multi-layered 3D structures containing a hiPSC-derived heterogeneous cell mixture (CMs and non-myocytes) in order to recapitulate disease-related phenotypes in vitro [59]. In addition, phase II clinical trials have been performed by Japanese scientists to evaluate the efficacy and safety of autologous skeletal myoblast sheet transplantation in patients with advanced heart failure. They demonstrated that the transplantation of engineered tissue promoted left ventricular remodeling, improved the heart failure symptoms and prevented cardiac death with a 2 year follow-up period [110]. Also, the potential use of cell-sheets that contain allogeneic hiPSC-CMs for clinical transplantation is under investigation [111].

A clear advantage of cell sheet technology for therapeutic applications is the absence of biomaterials, which reduces the risk of immune rejection that could arise from xenobiotic or non-autologous materials, and that no suture is needed to ligate the construct to the injured heart. Moreover, sheet technology enables direct cell-to-cell communications between cells in the transplanted sheets and the host tissue, facilitating electrical communication and vascular network formation within the cell sheet structure. On the downside, it has been argued that the fragility of these sheets makes them difficult to manipulate during implantation onto the heart [112]. Although cardiac tissue sheets have many advantages over other tissue engineering methods, these structures are not thick enough to reproduce the high complexity of the native myocardial tissue.

### 3.2. Scaffolds

ECT constructs made by repopulating cell-free scaffolds with suitable cells are usually referred to as cardiac patches. Scaffolds for CTE usually consist on a 3D polymeric porous structure that contributes to cell attachment and leads to the desirable cell interaction for further tissue formation [11,23,25,26,27,69,113,114,115]. Many different materials have been tested for the fabrication of scaffolds suitable for CTE, including natural and synthetic biomaterials. A commonly used synthetic material is polylactic acid (PLA), which is easily degradable forming lactic acid. PLA scaffolds were tested in some cardiovascular studies [116,117]. Another example is polyglycolic acid (PGA) and its copolymer with PLA poly(D,L-lactic-*co*-glycolic acid) (PLGA), an FDA approved biomaterial among the first tested for CTE due to its high porosity, biodegradability and processability [69,70,118]. However, it has been noted that the high stiffness of PLGA may limit the capacity of CMs to remodel the scaffold and ultimately impinge on their maturation [112]. Collagen, being the most abundant protein in the cardiac extracellular matrix (ECM), has a fibrillar structure that facilitates CM scaffolding. In addition to good biocompatibility, biodegradability and permeability, collagen also elicits low immunogenicity and can be engineered in various formats including high porosity scaffolds, all of which make of collagen the most commonly used biomaterial in scaffold-based CTE [21,22,23,24,25,28,119]. Other natural biomaterials used in this context include alginate, a polysaccharide derived from algae used in ECT constructs [26,120] due its high biocompatibility and appropriate chemical and mechanical properties [121], and albumin fibers, which have been used to create biocompatible scaffolds of high porosity and elasticity [29,122].

The combination of different approaches has enabled the development of scaffold systems of increasing complexity, thus bringing them morphologically closer to heart tissue. For example, researchers have generated fibrous scaffold with spatially distributed cues that enabled CM alignment within the patch [29,123,124]. However, major limitations of CTE still remain the generation of thick constructs (over ~100 µm in thickness) and the lack of electromechanical coupling between the cardiac patch and the host myocardium [100,112]. The generation of ECT constructs with a clinically relevant size requires ensuring that appropriate levels of oxygen and nutrients are maintained within the construct to satisfy the metabolic demand of CMs. Perfusion bioreactor systems pioneered by the Vunjak-Novakovic laboratory have proven to be of great value for the generation of thick ECT constructs full of viable cells with aerobic metabolism. In this case, cells were seeded and cultured in porous collagen scaffolds (11 mm in diameter, 1.5 mm in thickness) under continuous perfusion for 7 to 14 days, which led to the formation of contractile thick cardiac tissues [113,125]. More complex bioreactor systems designed to perfuse ECT constructs while also delivering electrical signals mimicking those in the native heart have also been developed [26,27,32]. Maidhof et al. used NRCMs seeded under perfusion into porous poly(glycerol sebacate) (PGS) scaffolds (8 mm in diameter, 1 mm in thickness), which were maintained under continuous perfusion at a flow rate of 18 µL/min and electrically stimulated at a frequency of 3 Hz. After 8 days, the combination of perfusion and electrical stimulation resulted in cell elongation, structural organization and improved contractility of ECT constructs [27]. Recently, our laboratories have reported the generation of 3D engineered thick human cardiac macrotissues (CardioSlices). Human iPSC-CMs were seeded together with human FBs into large 3D porous collagen/elastin scaffolds and cultured under perfusion and electrical stimulation in a custom-made bioreactor. Two weeks after culture, stimulated ECT constructs exhibited contractile and electrophysiological properties close to those of working human myocardium [32].

In addition to scaffolds made from synthetic or natural biomaterials, the use of matrices obtained by decellularizing native tissues has gained popularity for CTE. The process of decellularization allows obtaining natural ECM that can be used to mimic the native tissue structure. In essence, decellularized ECM could be recellularized with CMs or mixtures of CMs and other cell types, or with PSCs that would be differentiated in situ toward the desired cell types [126]. Tissues from a wide variety of sources including human, animals and plants have been used for this purpose [112]. The porcine heart is a prime example of tissue source for animal-derived decellularized scaffolds, due to its large size and to it being a preferred experimental model for cardiovascular research. In this case, it has been reported that the decellularization procedure allows obtaining a cardiac scaffold with preserved vasculature, mechanical integrity and biocompatibility [127]. Nevertheless, limitations noted with this approach include the extent of preservation of the ECM composition (which can be altered by the decellularization process), the difficulty in recellularizing the scaffold with clinically relevant numbers of CMs (in the order of billions), and the risk of eliciting immune rejection [112]. The issue of immune intolerance of animal-derived decellularized scaffolds has prompted research on plant-derived biomaterials as a source for ECT constructs [128]. Even though promising results have been reported using biomaterials derived from decellularized apple [129,130], spinach and parsley leaves [131], along with other cellulose-based scaffolds [132,133], further evaluation will be necessary to assess the usefulness of this type of materials for in vitro bioengineering and in vivo therapeutic applications.

Alternatively, human tissue might be a more appropriate source for decellularized ECM for therapeutic purposes, as it would address some of the limitations of animal- and/or plant-sourced materials described above [134,135,136,137]. In this respect, studies by Sanchez et al. demonstrated that the human acellular heart matrix can serve as a biocompatible scaffold for recellularization with parenchymal and vascular cells [138]. Moreover, Guyete et al. also used human decellularized heart tissue, which was in this case recellularized with iPSC-CMs and maintained in a custom-made bioreactor that provided coronary perfusion and mechanical stimulation. After 14 days in culture, the recellularized cardiac segment presented high metabolic activity and contractile function but exhibited low maturation state [139].

### 3.3. Hydrogels 

Embedding suitable cells in hydrogels provide important 3D information cues and, in the context of CTE, the constructs generated in manner are typically known as cardiac grafts. Hydrogels are among the most widely studied types of biomaterials in CTE (see Table 1). In particular, hydrogel-based materials have been shown to provide structural/mechanical support to cells [140], promote vascularization [141] and cell migration, differentiation and proliferation [142], and to improve cardiac function after implantation in murine and porcine models of MI [30,45,103]. Hydrogels can be made from different biomaterials that are usually classified into three types: natural (type I collagen, fibrin, gelatin, alginate, keratin, among other), synthetic [polycaprolactone (PCL), polyethylene-glycol (PEG), PLA, PGA and their co-polymer PLGA], and hybrid hydrogels, which are made by combining natural and synthetic polymers [143,144]. Natural-based hydrogels are often preferred for generating ECT constructs because of their high bioactivity, biocompatibility and biodegradability [142].

Cardiac “bundles” are the most common structures generated when using hydrogel-based systems and are cylindrical ECT constructs in the form of cables, ribbons or rings [112]. These structures are usually formed by embedding CMs from various sources within hydrogels made up of fibrin, type I collagen or other biomaterials, and maintaining them in culture until constructs become spontaneously contractile. The formation of these bundles results in self-alignment and anisotropic organization of CMs, which is a hallmark of cell maturation. Moreover, these constructs provide an easy way to analyze the electrical and mechanical properties of CMs, thus enabling the readily evaluation of their maturation state and facilitating their use in drug screening and toxicity assessment platforms [16,36,40,48,50,51,55,145,146,147].

In a pioneering approach, Eschenhagen and Zimmermann generated cardiac bundles (which they termed engineered heart tissues, or EHTs) by casting a mixture of NRCMs and a blend of type I collagen type I and Matrigel into cylindrical molds. Under conditions of mechanical stretching, the resulting ring-shaped constructs exhibited improved contractile function and a high degree of cardiac myocyte differentiation [33]. In subsequent work, five of such rings were stacked on a custom-made structure creating multiloop tissue constructs that survived after implantation and improved the cardiac function of infarcted rats [103]. Using the same principle, Kensah et al. produced cardiac bundles by seeding NRCMs with FBs in a collagen/Matrigel hydrogel into a Teflon casting mold between two titanium rods and subjected to mechanical and/or chemical stimulation [34]. Similarly, human ECT constructs have also been generated by casting a cell/hydrogel suspension in different types of molds between or around flexible posts. Schaaf et al. used hESC-CMs in a fibrin hydrogel seeded into an agarose casting mold between 2 elastic silicone posts for 5 weeks [37]. Controlling the 3D microenvironment has been further reported to induce spatial organization and promote CM maturation in hydrogel-based systems. In a study by Thavandiran et al., hESC-CMs and hESC-derived cardiac FBs were seeded in a collagen/Matrigel hydrogel into polydimethylsiloxane (PDMS) microwells with integrated posts. The authors compared two-well designs side by side: an elongated microwell containing posts in both extremes (capable of inducing uniaxial mechanical stress) and a square well containing posts around the edges (thus effecting biaxial mechanical conditioning). These studies demonstrated that constructs on elongated microwells showed comparatively better aligned sarcomeres and more elongated and longitudinally oriented CMs [38]. In turn, the Bursac laboratory created thin (~70 µm in thickness) 3D sheet-like constructs of large surface dimensions (7 × 7 mm) by casting hESC-derived CMs in fibrin hydrogels into PDMS molds with hexagonal posts, resulting in improved maturation at the functional (conduction velocities of up to 25 cm/s and contractile forces and stresses of 3.0 mN and 11.8 mN/mm^2^, respectively) and structural (increased sarcomeric organization and expression of cardiac genes) level [100]. Similarly, Turnbull et al. generated human ECT constructs with hESC-derived cells mixed in a collagen/Matrigel hydrogel in rectangular PDMS casting molds with integrated posts at each end and removable inserts. Forty-eight hours after casting, the inserts were removed from the mold, allowing the self-assembly of the tissues between the two flexible posts, which were used as force sensors. The resulting tissues exhibited typical features of human newborn myocardium tissue including contractile, structural and molecular characteristics [42].

Similar to hESC-CMs, iPSC-CMs also have been successfully cultured in hydrogel-based structures [11,40,46,47,49,50,51,53,55] (see Table 1). The Radisic laboratory pioneered the use of hiPSC-CMs to generate human ECT constructs by developing a platform in which cells in a collagen hydrogel organized around a surgical suture in a PDMS channel. The resulting 3D microstructures (3 mm^2^) were termed Biowires and contained aligned CMs that exhibited well-developed striations and showed improved cardiac tissue function after electrical stimulation [40]. A further improvement to this platform was the use of a polytetrafluoroethylene tube that allowed perfusion of the ECT microstructures and facilitated their use for drug toxicity testing [41]. More importantly, three independent studies reported in 2017 on the generation of clinical-size cardiac tissues by using hydrogel-based systems and hiPSC-CMs. Shadrin et al. generated human cardiac tissues of 36 × 36 mm that showed electromechanical properties close to those of working myocardium (conduction velocity of 30 cm/s and specific forces of 20 mN/mm^2^) by seeding hiPSC-CMs in a fibrin hydrogel into square PDMS molds [11]. Using a mixture of type I collagen and Matrigel with hiPSC-derived CMs and endothelial and vascular cells (in a 3:1:1 ratio), Nakane et al. generated rectangular ECT constructs with different shapes (bundles and mesh junctions, parallel bundles, plain sheets) and sizes (from 15 × 15 mm to 30 × 30 mm). They analyzed the association of CM orientation and survival with construct architecture and found that bundles and mesh junctions resulted in the highest myofiber alignment and lowest percentage of dead cells. Moreover, functional integration was observed after 4 weeks of transplantation onto rat uninjured hearts [50]. Large ECT constructs (35 × 34 mm) were also generated in the Zimmermann laboratory by seeding hiPSC-CMs and FBs in a collagen hydrogel on a 3D-printed construct holder with flexible poles in a hexagonal casting mold [51]. In a further refinement of this approach, Gao et al. generated human ECT constructs of 4 × 2 cm comprising hiPSC-derived CMs, SMCs, and ECs (3:1:1 ratio) in a fibrin hydrogel and maintained them in culture on a rocking platform. After 7 days of culture, the constructs showed improved electromechanical coupling, calcium-handling, and force generation [53]. 

Synthetic hydrogels have received comparatively less attention for CTE than those of natural origin. Ma et al. used PEG to create cardiac microchambers (100 to 300 µm in height) that induced spatial organization of hESCs and hiPSCs and directed their cardiac differentiation [43,44]. In addition, hybrid hydrogels have a great potential for CTE as they can mimic biological properties of the ECM and, at the same time, be tuned to suit the mechanical properties expected or desired for cardiac constructs [148]. Despite their great potential, much research is still needed to ascertain the specific advantages that synthetic and hybrid hydrogels may have over commonly used natural hydrogels in the context of CTE. At any rate, irrespective of the type of hydrogel used, current 3D cardiac grafts are constrained in maximum thickness by the ~300 µm limit of oxygen diffusion in passive culture systems and, therefore, while ideal for miniature structures with some tissue-like functionalities, they may not be suited for applications that require fully capturing the high complexity of the native heart tissue structure.

### 3.4. Cardiac Spheroids (and ‘Organoids’)

Spheroids are scaffold-free 3D cell constructs that rely on cell aggregation or self-organization and simulate aspects of the native cell microenvironment [149]. Cardiac spheroids can be constructed with CMs [150,151] and also include other cardiac resident cells such as FBs [152] or ECs [104]. Different percentages of various cell types have been tested for the generation of cardiac spheroids [153,154]. Spheroid-based systems are attractive to scientists for studying heterocellular interactions and drug effects because they only need low cell numbers to be formed. On the other hand, the absence of functional architecture limits the physiological analyses of the cells, like force generation and electrical conduction. Nevertheless, using the spheroid-based systems to deliver CMs into the damaged region of the heart has been reported. For instance, intramyocardial injection of cardiac spheroids in mice resulted in higher engraftment rates and improved electrical coupling with host myocardium, compared to single cell injection, which reveals potential for future clinical applications [155,156]. Moreover, several research groups are working on the generation of thicker functional structures using multicellular spheroids for further clinical testing. Noguchi et al. created a scaffold-free 3D tissue construct based on self-organization of 1 × 10^4^ spheroids. The individual spheroids were, in turn, obtained by combining 3 cell sources: NRCMs, hECs and hFBs in a 7:1.5: 1.5 ratio. ECT constructs generated in this way remained adherent and presented signs of vascularization seven days after transplantation onto the heart of nude rats [157]. In a related approach, scientists created a biomaterial-free cardiac tissue by 3D printing multicellular cardiac spheroids that displayed spontaneous beating and ventricular-like action potentials, which were engrafted as well into the rat heart tissue [158]. A final example of this approach is the study by Kim et al., who generated elongated 3D heterocellular microtissues by fusing together multicellular cardiac spheroids containing CMs and cardiac FBs. The authors demonstrated that such microtissues formed an electrical syncytium after seven hours in culture [159].

Similar to spheroids, organoids are scaffold-free 3D cell constructs that simulate aspects of the native environmental conditions. However, a critically important characteristic of organoids that sets them apart from spheroids, is that organoids contain organ-specific cell types that self-organize in a way that is architecturally similar to that of the native organ [160]. While some researchers may use the terms organoids and spheroids interchangeably, there are important differences between them. Both spheroids and organoids can be generated from PSCs, PSC derivatives, or tissue-specific stem/progenitor cells. When using PSCs, the technique of organoid formation is inspired in the embryoid body system [160], 3D aggregates of PSCs where cells undergo specification and differentiation into cell derivatives of the three main embryo germ layers [161]. In contrast, spheroids are much simpler than organoids in terms of the cell types that conform them, do not self-organize into organ-like patterns or structures, and depend to a lesser extent on ECM properties and composition [162]. Even though several published reports describe the generation of human “cardiac organoids”, these rely on direct cardiac differentiation of hiPSC-derived embryoid bodies [163] or aggregation of cardiac cell types (CMs, cardiac FBs and cardiac ECs) [164,165,166,167,168], which actually represent typical examples of spheroids [160,161]. We believe that the use of the term “cardiac organoids” in this context is misleading since the structures generated in those studies lacked the organ-like complexity characteristic of true organoids. Very recently, Lee et al. have described the generation of *bona fide* cardiac organoids in the mouse system that showed atrium- and ventricular-like structures highly reminiscent of the native embryonic heart. For this purpose, the authors induced sequential morphological changes in PSC-derived cells by including a laminin-entactin complex in the ECM and FGF-4 in serum-free medium [169]. Perhaps the application of similar approaches to hiPSC derivatives could lead to the generation of true human cardiac organoids containing relevant organ-specific cell types with the capacity to self-organize in organ-like structures. 

### 3.5. Heart-on-a-Chip

Microfluidic cell culture technologies enable researchers to create in vitro cell microenvironments that mimic organ-level physiology [170]. The term ‘organ-on-a-chip’ is generally applied to a microphysiological system, including the slides or plates that are connected to microfluidic devices to control perfusion of culture medium and exposure of defined stimuli [171]. Heart-on-a-chip technology refers specifically to microphysiological systems mimicking the function of heart tissue. In vivo-like cardiac cell culture systems could lead to a better understanding of (1) cardiac cell physiology; (2) cardiotoxicity of drugs intended for human used; (3) personalized treatments for CVD patients; and (3) mechanisms of heart regeneration [39,172,173]. In physiological conditions, the heart tissue is in direct contact with body fluids such as blood and lymph that exert physical forces (shear stress) on the cells. This continuous flow stimulation determines the cardiac cells structure, phenotype, intra- and extracellular interactions [174]. In vivo-like cardiac cell culture systems try to replicate these conditions in vitro. Thus, the heart-on-a-chip system provides suitable conditions to imitate biochemical, mechanical, and physical signals characteristic of heart tissue [175,176,177]. For example, it was shown that continuous perfusion enhances cell proliferation and parallel alignment of cells compared to static conditions [178]. In addition to perfusion, integrating mechanical and electrical stimulation into heart-on-a-chip devices also improves the maturation state of CMs [54,179,180,181], one of the key features for successful modeling of cardiac diseases [182]. Moreover, heart-on-a-chip systems are amenable to parallelization and thus to be used in high-throughput assays for drug screening and cardiotoxicity testing [173,183]. Particularly, the possibility of using hPSC-CMs brings an additional level of personalization to heart-on-a-chip systems [46,54,104,184,185,186,187]. For example, the Radisic laboratory developed a powerful platform, dubbed AngioChip, that integrated tissue engineering and organ-on-a-chip technologies to produce vascularized polymer-based microfluidic cardiac scaffolds. Such a platform can be used to generate both in vitro heart tissue models and in vivo implants for potential clinical application [186].

### 3.6. 3D Bioprinting

3D bioprinting is one of the latest additions to the tissue engineering toolbox, and one that could be used to create complex and large vascularized tissues [188,189]. Several methods of 3D bioprinting have been used in the context of CTE, including cell-laden hydrogel 3D structures [190], inkjet bioprinting [191], laser-assisted bioprinting [192], and extrusion-based bioprinting [193]. Biomaterials used in 3D bioprinting are based on piezo-resistive, high-conductance, and biocompatible soft materials. Gaetani et al. bioprinted a 2 × 2 cm ECT construct using human cardiac progenitor cells and alginate matrices, which was maintained for 2 weeks in vitro [120] or transplanted onto rat infarcted hearts where it led to cell engraftment [194]. Generation of 3D-bioprinted vascularized heart tissues using mouse iPSC-CMs with human ECs in a PEG/fibrin hydrogel has been recently reported showing improved connectivity to the host vasculature after subcutaneous transplantation in mice [195]. Despite the early stage of development, 3D bioprinting is a very promising technology for recapitulating the complex structure of heart tissue and already shows enormous potential in CTE. In a recent study, Noor et al. succeeded in bioprinting thick (2 mm) 3D vascularized and perfused ECT constructs that had high cell viability using an extrusion-based bioprinter. As bioinks, the authors used an ECM-based hydrogel derived from human decellularized omentum containing hiPSC-CMs, and gelatin containing iPSC-derived ECs and FBs. Computerized tomography of a patient’s heart was used to reproduce in vitro the structure and orientation of the blood vessels into the tissue. Bioprinted ECT constructs were transplanted into the omentum of rats and analyzed after seven days, when elongated and well-aligned CMs with massive striations were observed [56]. This study demonstrated that the possibility of using fully personalized materials makes 3D bioprinting technology very promising for clinical application by reducing the risk of immune rejection after transplantation. However, the system is still limited and further analyses should be performed to evaluate if heart tissue bioprinted in this manner could sustain normal blood pressure levels after transplantation [56].

3D bioprinting can also be combined with microfluidic systems to provide superior organ-level response with greater prediction of drug-induced capability [196]. On the other hand, recent advances in nanomaterial technology present an attractive platform for the creation of ECT constructs for biomedical applications. Electrospinning technology allows creating nanofibers with controlled dimensions and further development of 3D structures [197]. In addition, the nanofibrous structure provides appropriate conditions for pluripotent cells to anchor, migrate and differentiate [198]. Increasing attention is being given to these types of structures due to their distinct mechanical properties, high porosity and potential to induce formation of aligned tissues that can be successfully implanted to the heart [199].

### 3.7. Other Structures

So far, we have described the variety of approaches undertaken for the generation of ECT constructs that reproduce increasingly complex features of the human myocardial tissue. However, several groups have pursued approaches intended to model whole heart chambers. In an early attempt in 2007, scientists created a “pouch-like” single ventricle using a mixture of hydrogel composed of type I collagen and Matrigel and NRCMs, which was cultured in an agarose casting mold with the dimensions of a rat ventricle. After formation, it was transplanted onto the rat heart showing limited functional integration [200]. One year later, Lee et al. created a “cardiac organoid chamber” by seeding NRCMs mixed with a collagen/Matrigel hydrogel around a balloon-like shaped mold with a thickness of 0.65 mm and sizes of 4.5 to 9 mm for the unloaded outer diameter, equivalent to the sizes of embryonic rat hearts at 9.5–11 days of development. The authors succeeded in the creation of a heart ventricle but with a moderate contractile capacity [201]. In subsequent work, the same group used a similar method to produce a functional human cardiac chamber using hESC-CMs embedded in collagen-based hydrogel. They proved that such technology could facilitate drug discovery as it provides the capacity to measure clinically meaningful parameters of the heart like ejection fraction and developed pressure, as well as electrophysiological properties [202]. The Parker laboratory has also recently generated tissue-engineered ventricles by using nanofibrous scaffolds composed of PCL and gelatin seeded with either NRCMs or hiPSC-CMs on a rotating ellipsoidal collector. The authors showed that cells aligned following the fiber orientation, thus recapitulating the orientation of the native myocardium, although the contractile function was much smaller than those of the native rat/human ventricles most likely owing to the small thickness of the chamber [203]. Notably, the Feinberg group printed human cardiac ventricles in 2019 using the freeform embedding of suspended hydrogels (FRESH) method. In this case, type I collagen was used as a support material to create an ellipsoidal shell that was filled with hESC-CMs in a fibrinogen suspension. These engineered tissues displayed synchronized functional activity [204]. Finally, Kupfer et al. have very recently generated a 3D bioprinted chambered muscle pump using an optimized bioink formulation of ECM proteins that allowed hiPSC proliferation prior to differentiation. In this way, 3D bioprinted hiPSCs underwent cell expansion and differentiated into CMs in situ, yielding contiguous muscle walls of up to 500 μm in thickness. The resulting human chambered organoids showed macroscale contractions and continuous action potential propagation and were responsive to drugs and to external pacing [205].

## 4. Maturation of Engineered Cardiac Tissues

Regardless of the cell source or type chosen, biomaterial used, or method employed to generate ECT constructs, a critical aspect common to all of them is the requirement to promote or maintain the maturation state of CMs in culture. The application of biophysical stimuli mimicking those experienced by CMs in the native heart has been investigated in greatest detail and found to be among the most efficient ways to obtain highly mature ECT constructs. Thus, a variety of existing CTE technologies have incorporated the means to apply relevant biophysical stimuli during culture to promote cellular growth, orientation and/or maturation, and are reviewed below and summarized in Figure 1 and Figure 2.

### 4.1. Mechanical Stimulation

Mechanical cues are among the most important stimuli during heart development, ultimately determining the overall size and architecture of the organ, as well as regulating physiological postnatal changes in the heart resulting from the growth process, pregnancy, sports or injury [206]. Building on this notion, early studies investigated the importance of mechanical stimulation for CM maturation using embryonic chick cells [207] and NRCMs [60] embedded in collagen gels that were subjected to cyclic stretching. In later work, Zimmermann et al. used linear cyclic stretching to generate ECT constructs that exhibited contractile and electrophysiological properties similar to those of the rat native tissue. Specifically, the authors used circular molds to cast collagen-based constructs that, after 7 days in culture, were transferred to a device applying circular cyclic stretching. Transplantation of the constructs generated in this way to the healthy rat heart showed its fast integration and the high percentage of terminal cardiac differentiation achieved [33]. The positive role of mechanical stimulation for CM maturation was also reported by Kensah et al. [34]. In this case, cyclic mechanical stretching together with electrical stimulation of NRCMs resulted in improved maturation at the cell and tissue levels, which was proved by the increased cardiac gene expression and improved functional properties of the construct [34]. Similar results were obtained in another study where ECT constructs were subjected to mechanical compression and to the shear stress induced by medium perfusion in a bioreactor system. Additionally, elevated levels of bFGF and TGF-β and higher expression of Cx43 were observed under intermittent compression, which suggested improvement in cell-cell communication [28]. Moreover, combining mechanical stimulation with other strategies such as co-culturing CMs with ECs [36] or using highly defined media based on specific growth factors (FGF-2, IGF-1, TGF-β1, VEGF) [51] further increased the structural and functional maturation of ECT constructs derived from hESC and hiPSC [31,36,51]. Overall, the application of mechanical stimulation improves sarcomere organization, cardiomyocyte contractility, increases tensile stiffness and cell size.

On the other hand, the application of mechanical stimulation alone may not be enough for obtaining mature cardiac tissues. During heart development, CMs experience mechanical forces and electrical fields resulting from the synchronized contraction of the organ itself, which greatly influence their further differentiation and maturation [208,209]. For this reason, recent CTE studies used the combination of these two stimuli [35,38]. For example, Ruan et al. compared the influence of mechanical stimulation on the maturation of cardiac constructs alone or in combination with electrical stimulation. The authors created hiPSC-derived collagen-based 3D scaffolds that either underwent static stress conditioning for 2 weeks or were subjected to mechanical stimulation for one week followed by a second week with additional electrical stimulation. They demonstrated that, in response to mechanical loading alone, the engineered tissue showed increase in CM alignment and size, and improved contractility, force generation and passive stiffness of the constructs. The addition of electrical stimulation further improved force generation and increased expression of Ca-handling proteins but without further changes in cell size and alignment [47].

### 4.2. Electrical Stimulation

Electromechanical coupling is one of the major characteristics of mature CMs resulting in their synchronous response to the electrical pacing signals following with contractile function and pumping. The process of cardiac muscle contraction is generated by the depolarization of the CM membrane, which in the end leads to calcium release from the sarcoplasmic reticulum and subsequent contraction of the myofibrils [210]. Thus, researchers began to test the effects of electrical stimulation to train cardiac constructs. The Vunjak-Novakovic laboratory pioneered the application of electrical stimulation to ECT constructs by culturing NRCMs on thick collagen sponges and subjecting them to electrical pulses at 1 Hz for up to 8 days. They observed that electrical field stimulation induced cell alignment and improved ultrastructure organization along with an increase in the amplitude of synchronous contractions, whereas non-stimulated constructs displayed poorly developed ultrastructural features [23]. For the purpose of electrical stimulation, the group initially developed a simple electrical stimulation chamber with two carbon rod electrodes placed lengthwise along the bottom of a Petri dish that also allowed to evaluate the functional properties of the constructs [25]. In further refinements, more complex systems were developed that better mimicked conditions in the native heart, including a bioreactor system that allowed simultaneous perfusion and electrical stimulation of the ECT constructs [27]. The benefits of this combined system were highlighted by the presence of elongated CMs with marked striations, a well-defined intracellular structure, high levels of expression and proper localization Cx43, and increased conduction velocity and enhanced contractility of the construct. The combination of perfusion and continuous electrical stimulation has been extensively used by other research groups to promote functional maturity. For instance, Barash et al. integrated custom-made stimulating electrodes in a bioreactor system to electrically stimulate thick rat ECT constructs under perfusion. After 4 days in culture in this system, the constructs showed enhanced levels of Cx43 and improved cell striation and elongation [26]. In another study, Xiao et al. designed a microfabricated bioreactor with integrated electrodes to engineer mature rat microtissues for assessing in vitro pharmacological effects [41].

The positive effect of electric stimulation on CM maturation has been validated for human cells using hESC- and hiPSC-derived ECT constructs. Nunes et al. were the first to use a daily step-up protocol to test the effect of increasing stimulation frequency on human 3D self-assembled cardiac bundles (Biowires). Specifically, ramping up stimulation frequency over a 1-week period enhanced maturation, with a 1 to 6 Hz paradigm resulting in better results than a 1 to 3 Hz, as judged by improved sarcomeric organization, increased number of desmosomes and subsequent higher conduction velocities, and overall more mature Ca-handling and electrophysiological properties [40]. More recently, Ronaldson-Bouchard et al. generated ECT constructs of early-stage hiPSC-CMs (immediately after the initiation of spontaneous contraction) in fibrin hydrogels and maintained them in culture for one month while increasing the frequency of electric stimulation from 2 Hz to 6 Hz at a rate of 0.33 Hz per day. Tissues cultured under these conditions displayed a remarkably advanced state of maturation including adult-like cardiac gene expression and tissue ultrastructure, a positive force-frequency relationship, and functional calcium handling [55].

While the effects of electric stimulation have been analyzed in detail on the maturation of human ECT microconstructs and of perfused rat macroconstructs, perfusion bioreactors incorporating electrical stimulation have been more rarely employed to generate human ECT macroconstructs [43,51,52]. Indeed, only recently have hiPSC-derived cells and 3D thick scaffolds been combined with continuous electrical stimulation to develop macrotissues of human CMs. For this purpose, the group of one the authors developed a parallelized perfusion bioreactor with custom-made culture chambers endowed with electrostimulation capabilities. We showed that hiPSC-CMs readily survive and mature after long-term culture within large (10 mm in diameter by 2 mm in thickness) 3D porous scaffolds, giving rise to macroscopically contractile tissues, which we termed ‘CardioSlices’. More importantly, we found that continuous electrical stimulation of these large ECT constructs for 2 weeks promoted the alignment and synchronization of CMs, and the emergence of complex cardiac tissue-like behaviors, including the spontaneous generation of electrophysiological signals strikingly similar to human electrocardiograms, which could be readily registered from the construct surface, and a response to proarrhythmic drugs that was predictive of their effect in humans [32].

### 4.3. Coculture with Non-CMs

Since cardiac cells other than CMs play important roles in heart homeostasis, several groups have explored whether their incorporation into ECT constructs improved cardiac maturation. Indeed, epicardial cells closely interact with CMs and regulate myocardial wall development [211] and, through paracrine mechanisms, influence heart electrophysiology [212,213]. Cardiac endocardial cells are found in direct contact with CMs, supporting their metabolic activity and regulating their contractility via paracrine signals [214]. Studies on neonatal and fetal CMs have shown that inclusion of cardiac FBs and/or ECs into ECT constructs improved the morphological and functional maturation of CMs [113,215,216]. Thavandiran et al. directly tested whether the cell composition of ECT constructs affected their functionality. For this purpose, the authors incorporated ESC-derived CMs and CFs in different ratios into collagen based microtissues, and found that the 3:1 (CMs:CFs) ratio provided the best microenvironment based on increased expression of cardiac maturation genes [38]. Similarly, Gao et al. generated a clinical size human ECT construct using hiPSC-CMs, -ECs and -SMC (2:1:1) in a fibrin hydrogel and reported a significant functional maturation of the construct in vitro and high engraftment rates in vivo [53]. Thus, the combination of more than one cell type within the ECT constructs can also aid in promoting their morphological and functional maturation [36,51,58,59]. 

## 5. Functional Assessment of Engineered Cardiac Constructs

The development of CTE approaches to increase the cardiac maturity of ECT constructs is necessarily mirrored by advances in the methodologies used to evaluate their degree of maturation. As described in the previous sections, the vast majority of human ECT constructs developed to date are hydrogel-based and do achieve measurable increases in CM maturation when analyzed at the cellular level, both in miniature constructs and thin sheets. Specific improvements in CM maturation were most frequently analyzed in terms of gene/protein expression profile, sarcomeric structure and/or cell electrophysiological properties being closer to those of adult CMs. Some studies also analyzed functional tissue-like properties of the engineered constructs, such as force generation and contractility, and a few studies also measured propagation of the electrical signals by analyzing calcium transients or voltage-sensitive dyes (see Table 1). The advantages and limitations of the technologies used to assess the functional maturation of ECT constructs are summarized in Table 2 and discussed below.

### 5.1. Patch-Clamp and Microelectrodes

Patch-clamp is the gold standard technique used to study the electrophysiological properties of excitable single cells such as PSC-CMs [37,40,49,51,55]. This technique allows measuring transmembrane voltage and currents using pulled glass micropipettes [222]). In addition, electrical coupling between two cells can be measured by applying patch electrodes to each of them separately [223]. Changes in membrane potential like action potentials can be measured by a whole-cell current clamp or using sharp microelectrodes with narrow bores [33,49,224], which allow precise registering of action potentials due to preservation of the true intracellular milieu [217]. While ideal for recording isolated CMs, patch-clamp technology presents important limitations for the study of ECT constructs. In addition to the inherent low throughput of the technique and the need for specialized equipment and highly trained personnel, patch-clamp requires direct visualization of the cell(s) being analyzed, which is not straightforward when cells are inside an ECT construct. Moreover, patch-clamp measures the electrophysiological properties of CMs themselves, rather than those of the construct as a tissue.

Functional assays used to evaluate the cardiac maturation state of the constructs, both at the cellular and the tissue levels, indicating the main advantages and disadvantages of each approach, as well as references to publications where those approaches were used in the context of CTE. CM, cardiomyocyte; ECT, engineered cardiac tissue. 

### 5.2. Multielectrode Arrays

Multielectrode arrays (MEAs) are devices that contain arrays of electrodes distributed over a small surface on which cells can be cultured directly. MEAs are generally used for millimeter-scale analysis of extracellular potential within beating clusters of CMs [43,225], where they can measure extracellular field potential, spontaneous beating rate, repolarization characteristics, and conduction velocity and trajectories [226]. Unlike patch-clamp, MEA technology can be used to assess the electrophysiological properties of CMs at both the single-cell and tissue levels in cell monolayers or ECT constructs [59,108,218], and allows long-term recording of electrophysiological parameters noninvasively [219]. The main limitations of MEA technology for the functional evaluation of ECT constructs are that measurements are restricted to the small areas detected by the electrodes, and that it provides incomplete information about the shape of the propagated action potentials [217].

### 5.3. Optical Mapping

Optical mapping is a non-invasive method for the assessment of electrical activity that is well established for the study of cell monolayers [227] and has also been applied to ECT constructs [29,35,40,48,55,99,118,220]. This technique relies on the use of voltage-sensitive (di-4-ANEPPS, di-8-ANEPPS and RH237) [228] or Ca2+-sensitive fluorescent dyes (rhod-2 and fluo-4). The main advantage of this technique is the ability to monitor the electrical activity and shape of action potentials from many areas of the construct. On the other hand, these dyes have cytotoxic effects that limit the maximum duration of experiments [221]. Moreover, the dyes are light-sensitive and subject to photobleaching, further limiting the time of analysis and reassessment of the sample. Optical mapping has been combined with MEA technology for the study of conduction velocity, direction of the signal and visualization of arrhythmias [229].

### 5.4. Force Transducers

The mechanical force exerted by contracting CMs can be measured in single cells as well as in whole ECT constructs [36,42,47,48,50,57,100]. For single-cell measurement, one end of the CM is attached to a rigid steel needle and the other end to a piezo-electric motor for its stretching. Force measurements are then taken by a sensitive force transducer connected to the needle [230]. For tissue-level properties, the whole construct is connected to a force transducer [36,47,48,100]. Here, the main disadvantage is the inability to measure the contractile stress from each CM. Advantages, on the other hand, include the larger magnitude of the signals obtained and the possibility to measure the therapeutic potential of ECT constructs before implantation [217].

To date, those are the most complex multicellular (tissue-level) behaviors analyzed in the context of CTE. New methods with greater sensitivity will be needed for evaluating tissue-like functional properties. Recently, the group of one the authors has explored the use of surface electrodes to record the electrical activity of thick ECT constructs. The bioelectrical signal detected in this way was extremely informative of the activity of electrically-active cells within the construct and, most importantly, of the intercellular coupling at the macro (whole construct) scale, resulting in complex waves very reminiscent of surface ECG recording in humans. In our study, we showed that electrical stimulation of the constructs resulted in improvements in tissue-like properties (shape of ECG-like signals and response to drugs) that could not be predicted by the small increases in maturation achieved at the CM level [32].

## 6. Summary and Future Directions

One of the most pressing current challenges in CTE is how to achieve a level of cardiac maturation that resembles that of the adult human myocardium. This is even more important for experimental approaches based on PSC-CMs that, while clearly taking the center stage of CTE, are notoriously difficult to mature ex vivo [182,231]. The use of immature cells/constructs could lead to the improper interpretation of in vitro drug testing results and inaccurate prediction of their effect in vivo upon transplantation. We have reviewed here the variety of imaginative approaches researchers have employed to obtain mature CET constructs using mechanical and/or electrical stimulation and coculture with non-myocyte cells, as well as biochemical interventions. These approaches have proven successful in promoting CM maturation from an embryonic-like state to cells displaying late fetal phenotypes. We have also reviewed how transplantation of such CET constructs had positive impact in cardiac function in experimental animal models and even patients. However, it should be noted that, to our knowledge, no approach has yet succeeded in obtaining CMs with maturity features beyond those of early postnatal CMs. Moreover, much of the earlier work attempting to obtain adult-like cardiac constructs concentrated on increasing the degree of maturation at the CM cell level, and had limited success in achieving functional tissue-like properties [40,47,51,55]. Taking into account that the myocardial tissue is a complex system that requires the synchronized operation of CMs as a functional syncytium [232], we contend that improving high-order behaviors at the macroscale (tissue) level could lead to more adult-like phenotypes, even when the maturation state of individual CMs were only marginally improved [32,47]. Future research will be necessary to explore whether the synergistic exploitation of approaches improving cardiac maturation at both the CM and the tissue levels will result in ECT constructs with the features of adult human myocardium. Moreover, pending issues such as improving the functional integration of ECT constructs within the host myocardium, increasing their durability and scalability, along with making them amenable to cryopreservation, surely warrant further research. Considering the multidisciplinary nature of CTE, further developments in the field will require the coordinated efforts from researchers with diverse backgrounds.

## Figures and Tables

**Figure 1 ijms-22-01479-f001:**
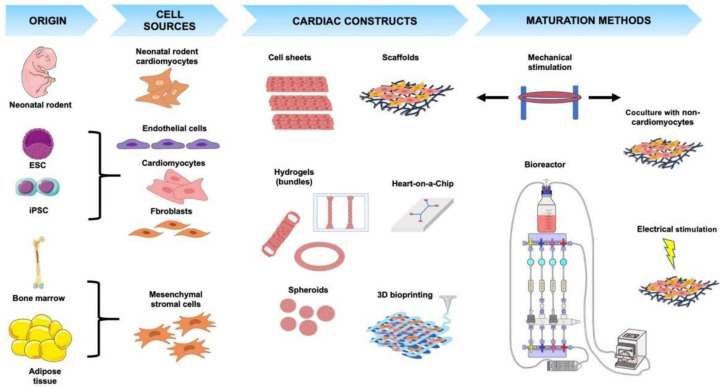
Main types of cell sources, cardiac constructs and maturation methods used in CTE. The cell types most frequently used as source for CTE include neonatal rodent cardiomyocytes, ESC-or iPSC-derived endothelial cells, cardiomyocytes and fibroblasts, and mesenchymal stromal cells derived from bone marrow or adipose tissue. Different systems have been developed to generate ECT including cell sheets, scaffolds, hydrogels, bundles, spheroids, heart-on-a-chip, and 3D-bioprinted constructs. The main methods used to increase maturation of cardiac constructs comprise mechanical and/or electrical stimulation, co-culturing with non-myocyte cells and continuous perfusion within bioreactors. See main text for details. Figure created with BioRender.com and smart.servier.com.

**Figure 2 ijms-22-01479-f002:**
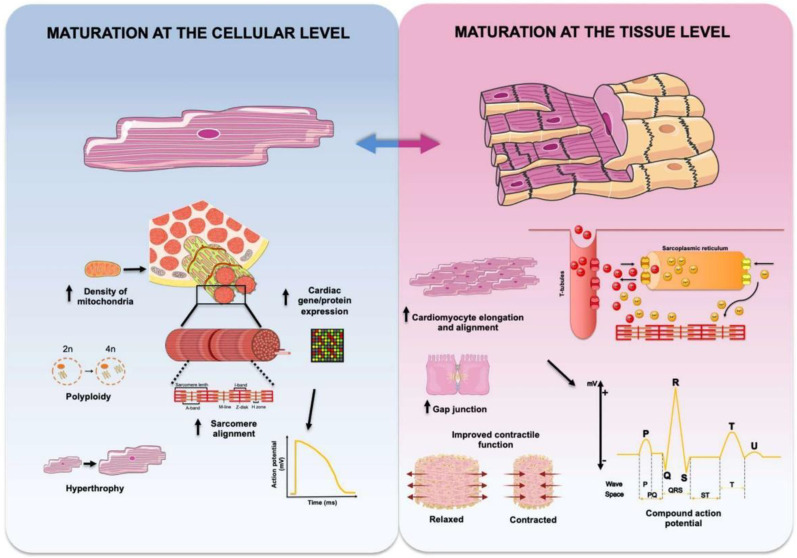
General features of cardiac maturation at the cellular and tissue levels. Cardiac maturation at the level of cardiomyocytes (blue-shaded background) can be experimentally ascertained by a more adult-like cardiac gene expression profile, changes in sarcomeric ultrastructure resulting in sarcomere elongation and alignment, increased mitochondrial density to accommodate for higher energy demand, and massive growth in cardiomyocyte size (hypertrophy) with increased percentage of polyploid cells. Electrophysiologically, cardiomyocyte maturation results in a more negative resting membrane potential and increased duration and amplitude of action potential. Cardiac maturation at the tissue level (pink-shaded background) is characterized by elongated cardiomyocytes preferentially aligned along a specific axis, increased expression of Cx43 and distinct localization of gap junction proteins, and improvements in impulse propagation and synchronization of cardiac contraction, leading to cardiomyocytes performing as a functional syncytium. Maturation at the tissue level results in increased contractility and force generation and, electrophysiologically can be registered as compound action potentials reminiscent of surface ECG recordings. Figure created with BioRender.com and smart.servier.com.

**Table 1 ijms-22-01479-t001:** Representative studies on CTE.

CTE Construct			Cell Source		Culture Conditions			Functional Analyses		Ref.
Type	Area (mm^2^)	Thickness (mm)	Species	Cell Type	Stimulation	Perfusion	Time (d)	CM Maturation	Tissue Maturation	
Scaffold	70	2	Rat	Neonatal CM	No	Yes	10	Protein expression	No functional analyses	[21]
Scaffold	95	1.5	Rat	Neonatal CM	No	Yes	7	Protein expression	Contractile activity	[22]
Scaffold	48	1.5	Rat	Neonatal CM	Electrical stimulation	No	8	Protein expression, sarcomere structure	Contractile activity	[23]
Scaffold	95	1.5	Rat	Neonatal CM	No	Yes	14	No functional analyses	Contractile activity	[24]
Scaffold	48	1.5	Rat	Neonatal CM	Electrical stimulation	No	8	Protein expression, sarcomere structure	Contractile activity	[25]
Scaffold	20	2	Rat	Neonatal CM	Electrical stimulation	Yes	4	Protein expression	No functional analyses	[26]
Scaffold	50	1	Rat	Neonatal CM	Electrical stimulation	Yes	8	No functional analyses	Contractile activity	[27]
Scaffold	20	2	Rat	Neonatal CM	Mechanical stimulation	Yes	4	Protein expression	No functional analyses	[28]
Scaffold	80	5	Rat	Neonatal CM	No	No	7	No functional analyses	Calcium imaging	[29]
Scaffold	n/a	n/a	Human	ESC-CM	Mechanical stimulation	No	14	Protein expression	Force of contraction	[30]
Scaffold	56	3.5	Human	ESC-CM	Mechanical	No	5	Gene/Protein expression	Calcium imaging	[31]
Scaffold	80	1	Rat/human	Neonatal CM/iPSC-CM	Electrical stimulation	Yes	14	Gene/protein expression, sarcomere structure	Force of contraction, contractile activity, whole construct electrical activity	[32]
Hydrogel	n/a	n/a	Rat	Neonatal CM	Mechanical stimulation	No	14	Protein expression, sarcomere structure, electrical signal propagation	Force of contraction	[33]
Hydrogel	n/a	0.9	Rat	Neonatal CM	Mechanical stimulation	Yes	14	Gene/protein expression	Force of contraction	[34]
Hydrogel	n/a	n/a	Rat	Neonatal CM	Electro-mechanical stimulation	No	13	Protein expression	Force of contraction, calcium imaging	[35]
Hydrogel	60	n/a	Human	ESC-CM/iPSC-CM+HUVEC+MSC	Mechanical stimulation	No	4	Gene expression, sarcomere structure	Force of contraction	[36]
Hydrogel	0.4	n/a	Human	ESC-CM	No	No	14	Gene/protein expression, patch clamp	Force of contraction	[37]
Hydrogel	n/a	0.1	Rat/human	Neonatal CM/ESC-CM	Electro-mechanical stimulation	No	7	Gene/protein expression	Contractile activity, optical mapping	[38]
Hydrogel	49	n/a	Human	ESC-CM	No	No	14	Gene/protein expression, sarcomere structure	Force of contraction, optical mapping	[39]
Hydrogel	3	0.3	Human	iPSC-CM	Electrical stimulation	No	14	Protein expression, sarcomere structure, patch clamp	Contractile activity, optical mapping, calcium imaging	[40]
Hydrogel	15	n/a	Rat/human	Neonatal CM/ESC-CM	Electrical stimulation	Yes	14	Protein expression	Force of contraction, contractile activity	[41]
Hydrogel	5	n/a	Human	ESC-CM	No	No	24	Gene/protein expression, sarcomere structure	Force of contraction, optical mapping	[42]
Hydrogel	n/a	n/a	Human	iPSC-CM	No	No	7	Protein expression, electrical signal propagation	No functional analyses	[43]
Hydrogel	0.125	n/a	Human	iPSC-CM	No	No	15	Gene/protein expression	No functional analyses	[44]
Hydrogel	27	0.2	Human	iPSC-CM+iPSC-EC/HUVEC	No	No	15	Gene/Protein expression	Optical mapping	[45]
Hydrogel	4	n/a	Human	iPSC-CM	No	No	40	Protein expression, sarcomere structure	Force of contraction	[46]
Hydrogel	20	0.3	Human	iPSC-CM	Electro-mechanical stimulation	No	14	Protein expression, sarcomere structure	Force of contraction, calcium imaging	[47]
Hydrogel	14	0.2	Rat/human	Neonatal CM/iPSC-CM	No	No	14	Gene/protein expression	Force of contraction, optical mapping	[48]
Hydrogel	200	n/a	Human	iPSC-CM	No	No	60	Protein expression, electrical signal propagation	No functional analyses	[49]
Hydrogel	900	n/a	Human	iPSC-CM	No	No	28	Protein expression	Force of contraction, contractile activity	[50]
Hydrogel	1190	0.5	Human	ESC-CM/iPSC-CM	Mechanical stimulation	No	45	Gene expression, sarcomere structure, patch clamp	Force of contraction	[51]
Hydrogel	1296	0.1	Human	iPSC-CM	No	No	21	Gene/protein expression, sarcomere structure	Force of contraction, optical mapping	[11]
Hydrogel	100	0.1	Rat/human	Neonatal CM/ESC-CM	Electrical stimulation	No	7	No functional analyses	Force of contraction, contractile activity	[52]
Hydrogel	800	n/a	Human	iPSC-CM+iPSC-EC+iPS-SMC	No	No	7	Protein expression	Force of contraction, optical mapping	[53]
Hydrogel	5	0.3	Human	ESC-CM/iPSC-CM+hcFB	Electrical stimulation	No	42	Gene/Protein expression, sarcomere structure, patch clamp	Force of contraction, contractile activity, calcium imaging	[54]
Hydrogel	11	n/a	Human	iPSC-CM	Electrical stimulation	No	30	Gene/protein expression, sarcomere structure, patch clamp	Force of contraction, calcium imaging	[55]
Hydrogel	14	2	Human/Pig/Rat	iPSC-CM+iPSC-EC/Neonatal CM+HUVEC+FB	No	No	7	Gene/Protein expression	Optical mapping, calcium imaging	[56]
Cell sheets	116	0.045	Rat	Neonatal CM	No	No	4	Protein expression, sarcomere structure, electrical signal propagation	Force of contraction	[57]
Cell sheets	960	0.1	Rat/human	Neonatal CM+EC	No	Yes	10	Protein expression	No functional analyses	[58]
Cell sheets	70	0.1	Human	iPSC-CM+MSC	No	No	4	Protein expression, electrical signal propagation	Contractile activity	[59]

**Table 2 ijms-22-01479-t002:** Main approaches to functional assessment of ECT constructs.

Approach	Advantages	Disadvantages	Ref.
*Patch-clamp*	- Direct recording of action potentials - Ideal for recording isolated CMs	- Requires specialized equipment/personnel - Requires direct cell visualization - Not well suited for ECT constructs	[217]
*Multielectrode arrays*	- Allows assessment of single CMs and ECT constructs - Noninvasive method. - Allows long-term recording	- Measurements restricted to small area of the ECT construct. - Incomplete information of action potential propagation	[59,108,217,218,219]
*Optical mapping*	- Allows assessment of CM monolayers and ECT constructs - Noninvasive method - Unsophisticated equipment	- Indirect measurement of electrophysiological parameters - Cytotoxicity of dyes may limit measurement duration and preclude sample reassessment	[29,35,39,48,55,118,220,221]
*Force transducers*	- Allows assessment of single CMs and ECT constructs - Allows measuring direct and indirect contractile force - Allows long-term measurements	- Unsuitable for assessing contractile stress from single CMs within ECT constructs	[36,39,42,47,48,50,57,217]

## Data Availability

Not applicable.

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
