# Peer review of "Engineering and Assessing Cardiac Tissue Complexity"

_ijms, 2021, doi:10.3390/ijms22031479_

Round 1

Reviewer 1 Report

The manuscript by K. Tadevosyan and colleagues shows advances in cardiac tissue engineering and its potential application to study disease mechanisms and therapeutics.

This review is well-organized and well-written.

Major comment:

1- In line with studies suggesting the effect of epicardial white adipose tissue on cardiac remodeling and function, the potential use of cardiac tissue engineering systems in the investigation of mechanisms of disease would be lacking, e.g., screening of cocultured explants/secretomes of diseased (i.e., diabetic) adipose tissue. Additionally, the impact of experimental therapies on cardiac function was not eventually mentioned. Maybe it could add value to this review or at least mentioned in the last subsection.

Minor comments:

1- Figure 1. “main types of cells” might be “main types of cell sources” should be indicated in Figure 1 caption. Also, the abbreviation CTE might be used instead of cardiac tissue engineering.

Reviewer 2 Report

Critical evaluation of the manuscript entitled Engineering and assessing cardiac tissue complexicity submitted by Tadevosyan et al.

General comments:

This manuscript consists of 30 numbered pages cited 245 references. The references are properly collected and arranged. In formal aspect it is an appropriate work, and meets the requirements determined by the journal.

Specific comments:

The objective of this manuscript is to give an overview of the general approaches developed to generate functional cardiac tissues. Different cell sources, biomaterials and the types of engineering strategies were discussed. This review will support the specialist who wants to engineer cardiac tissue constructs and using this work, they will be able to screen and select  the best suited preparation for each particular application.

Critical remarks:

The manuscript is well-written and the authors collected the references properly, but the table is a little bit caotic. The authors should rearrange it based on ie. the type of cardiac tissue engineering. More table should be prepared to understand the content of this review.

The summary is too short, and the authors should suggest clear direction toward the specialist in the summary.

Author Response

Please, see attachment.
